# Baseline knowledge and attitudes on COVID-19 among hotels' staff: A cross-sectional study in Kigali, Rwanda

Aphrodis Hagabimana[1,2]*, Jared Omolo[1], Ziad El-Khatib[3,4,5], Edson Rwagasore[6], Noella Benemariya[1], Olivier Nsekuye[1], Adeline Kabeja[6], Helene Balisanga[6], Angela Umutoni[6], Aimable Musafili[7], Albert Ndagijimana[1]

1 Rwanda Field Epidemiology and Laboratory Training Program, Department of Epidemiology and Biostatistics, University of Rwanda, Kigali, Rwanda, 2 Kigeme District Hospital, Nyamagabe District, Kirehe, Rwanda, 3 Bill and Joyce Cumming Institute of Global Health, University of Global Health Equity, Kigali, Rwanda, 4 Department of Global Public Health, Karolinska Institutet, Solna, Sweden, 5 World Health Programme, Université du Québec en Abitibi-Témiscamingue, Rouyn-Noranda, Canada, 6 Public Health Surveillance and Epidemic Response, Rwanda Biomedical Centre, Kigali, Rwanda, 7 Department of Pediatrics and Child Health, School of Medicine and Pharmacy, College of Medicine and Health Sciences, University of Rwanda, Kigali, Rwanda

* hagaba01@gmail.com

**Data Availability Statement:** All relevant data are within the manuscript.

## Abstract

### Background

The World Health Organization declared coronavirus disease 2019 (COVID-19) as a global pandemic on the 11th of March, 2020. Hotels and other public establishments have been associated with higher transmission rates. Sensitisation of staff and strengthening of Infection Prevention and Control (IPC) practices in such settings are important interventions. This study assessed the baseline knowledge and attitudes on COVID-19 among hotels' representatives in Kigali, Rwanda.

### Methods

A cross-sectional study was conducted among hotels' staff in Kigali in July 2020. A structured questionnaire was self-administered to 104 participants. Baseline knowledge and attitudes were assessed using a number of pre-test questions and mean scores were used to dichotomise the participants' responses as satisfactory or unsatisfactory.

### Results

All of the 104 hotels' staff completed the self-administered questionnaires. Sixty-seven percent (n = 70) were male and 58% (n = 60) were aged between 30 and 44 years. The satisfactory rate of correct answers was 63%±2.4 (n = 66) on knowledge and 68%±1.7 (n = 71) on attitudes evaluation. Participants with University education were more likely to have satisfactory knowledge (AOR: 2.6, 95% C.I: 1.07–6.58) than those with secondary education or less. The staff working in the front-office (AOR: 0.05; 95% CI 0.01–0.54) and housekeeping

**Funding:** The author(s) received no specific funding for this work

**Competing interests:** The authors have declared that no competing interests exist.

(AOR: 0.09; 95% C.I: 0.01–0.87) were less likely to have satisfactory attitudes than those working in the administration.

## Conclusions

Hotels' staff based in the capital of Rwanda have shown satisfactory knowledge and attitudes regarding appropriate IPC practices for preventing the COVID-19 transmission. Educational interventions are needed to improve their knowledge and attitudes for better prevention in this setting.

## Introduction

The Coronavirus disease 2019 (COVID-19) is caused by a novel coronavirus (SARS-CoV-2), which was reported for the first time in December 2019 to be transmitted from an animal to humans [1]. Currently, it spreads rapidly from a person to person via droplets or direct contact and touch of a contaminated surface. In addition, it was found in faecal samples and anal swabs of some patients [2]. The main clinical signs and symptoms of COVID-19 include cough, fever $\geq 38°C$, severe shortness of breath, dry cough and other clinical characteristics, including diarrhoea, vomiting and runny nose [3, 4].

The Rwanda Ministry of Health has strengthened the COVID-19 surveillance countrywide and different modes of technologies combined with prevention strategies were conceived to minimize the rate of transmission [5]. On the 14th of March 2020, the Government of Rwanda considered a total lockdown, including the hotel industry as a strategic measure to limit the national transmission of SARS-CoV-2 [6]. On the 4th of May 2020, Rwanda eased its total lockdown and allowed businesses and public transportation to resume but under well-defined health and safety measures. This included all re-opened public spaces mandated to have hand-washing stations in place, temperature monitoring devices, use of face masks, and practice of physical distancing of at least one meter [7].

The preventive measures, health education, and public awareness have shown an impact in addressing and controlling COVID-19 [7, 8]. However, hotel establishments are more susceptible to contagion than other public establishments because they are visited by many people who interact among themselves and with the staff. Therefore, hotels' staff are at higher risks for the aspects of lodging of guests, food and beverages handling and serving, cleaning, organisational activities and require specific attention for every member to strictly comply with the basic protective measures against COVID-19 [9]. This study assessed the baseline knowledge and attitudes of hotels' staff regarding appropriate IPC practices required to ensure their safety and protection of guests and customers, attending hotels located in Kigali, Rwanda, in the context of pandemic COVID-19.

## Materials and methods

### Study design and setting

This was a cross-sectional study conducted in July 2020 to assess the baseline knowledge and attitudes of hotels' staff in Rwandan capital, Kigali. This city is geographically located at the heart of Rwanda and it includes three districts with a total population of approximately 1.2 million inhabitants [10]. In 2012, the Rwanda Development Board (RDB) indicated that 50% of 486 hotels, which were counted countrywide, were located in Kigali [11].

## Study population

This study included a total of 52 hotels, which were the most visited in Kigali, as suggested by the RDB. These hotels had common areas, which could ensure social distancing of at least one meter between guests. Each selected hotel allocated two representatives to attend a training on COVID-19 transmission and prevention, which was planned by Rwanda Biomedical Centre (RBC) in collaboration with RDB. This training took place in July 2020 and was facilitated by trainers, who were previously trained by RBC on the subject. All trainees were considered eligible to the study.

## Data collection

The main investigator, also a trainer of participants, together with co-authors developed a structured questionnaire based on a core set of questions related to the knowledge and attitudes on COVID-19 transmission and prevention. These questions also included knowledge and attitudes on decontamination and waste management, as suggested by the World Health Organization and Rwandan national guidelines [12–14].

All questions were categorised into three groups. The first group included questions related to the socio-demographics characteristic of hotels' staff: age, gender, educational level, work experience, and job category, work category inside the hotel and the most reliable sources of information about COVID-19 (family members, friends, official websites, radio, television, and other social media). The second group included questions pertaining to the participants' disease awareness and clinical signs and symptoms of COVID-19. The third group comprised questions related to the COVID-19 prevention and control. Authors rechecked the questionnaire for accuracy and completeness prior the data collection.

## Data management and analysis

Descriptive statistics were performed and data were summarised into frequency tables. The levels of knowledge and attitudes of hotels' staff on COVID-19 were assessed using a numeric pattern of scoring, which was applied elsewhere [15]. This pattern assigned a score of "1" for the correct answer and "0" for the false answer and neutral for the unknown response. The levels of measured outcomes were expressed as mean scores with standard deviation (Mean ± SD). The cut-off points for satisfactory levels of knowledge and attitudes were set at greater than the mean scores. The unsatisfactory levels of knowledge and attitudes were considered less than the mean scores.

Cross-tabulations were applied to analyse the patterns of mean scores across various socio-demographic categories of participants. When analysing data, sociodemographic variables were considered as covariates. Logistic regression was used to analyse the association between covariates and the outcome, which was dichotomous: satisfactory or unsatisfactory mean score. The strength of association was assessed using unadjusted odd ratio (OR) in bivariate analysis and adjusted odd ratio (AOR) in multivariate analysis. The level of significance was considered as $p < 0.05$. Epi-Info (version March 2015, CDC Atlanta) was used for data entry and Stata version 13 (StataCorps.2013) for data analysis.

This study was done through the collaboration between Rwanda Biomedical Centre; National Infection, Prevention and Control COVID-19 Joint task force Coordination; and Rwanda Development Board. Verbal informed consent was sought from participants who were given the right to choose to respond to any question in the survey with a guaranteed confidentiality. The ethical approval was obtained from the Rwanda Public Health Surveillance and Emergency Preparedness and Response Division (Ref: No: 12/RBC/PHSEPR/2021).

## Results

The Table 1 shows sociodemographic characteristics of the study population. Out of 104 participants, 57.7% (n = 60) were aged between 30 and 44 years, 67.3% (n = 70) were male, 72.1% (n = 75) had University educational level, 51.9% (n = 54) had a working experience in hotels of 4 to 10 years, and 34.6% (n = 36) were working in food and beverage unities. Those who used social media as the most reliable source of information on COVID-19 were 29.8% (n = 31).

Knowledge and attitudes of participants regarding COVID-19 prevention and control were reported in the Table 2. Most participants, 52.9% (n = 55), were aware that COVID-19 was a fatal disease. The vast majority of them; 94.2% (n = 98), showed a positive attitudes regarding the handwashing practice using clean water and soap to prevent COVID-19 transmission.

As shown in Table 3, most participants had satisfactory rates of correct answers on knowledge and attitudes evaluation. The maximum scores were comparable on both knowledge and attitudes evaluation.

As shown in Table 4, the participants with a University educational level were more likely to have satisfactory knowledge rate (AOR: 2.6, 95% C.I: 1.07–6.58) than those with a secondary educational level or less. The staff working in the front-office (AOR: 0.05; 95% CI 0.01–0.54) and housekeeping (AOR: 0.09; 95% C.I: 0.01–0.87) were less likely to have satisfactory attitudes

**Table 1. Sociodemographic characteristics of hotels' staff in Kigali (N = 104).**

| Variables | Frequency | Percentage (%) |
|---|---|---|
| **Age** | | |
| 18–29 years | 35 | 33.6 |
| 30–44 years | 60 | 57.7 |
| 45–60 years | 7 | 6.7 |
| ≥61 years | 2 | 1.9 |
| **Gender** | | |
| Man | 70 | 67.3 |
| Woman | 34 | 32.7 |
| **Educational level** | | |
| Secondary | 29 | 27.9 |
| University | 75 | 72.1 |
| **Work experience duration (years)** | | |
| >1 years | 23 | 22.1 |
| 1–3 years | 18 | 17.3 |
| 4–10 years | 54 | 51.9 |
| 11 years and over | 9 | 8.6 |
| **Work category inside the hotel** | | |
| Administrative | 24 | 23.1 |
| Food and beverage | 36 | 34.6 |
| Front officer | 19 | 18.3 |
| Housekeeping | 25 | 24.0 |
| **Most reliable source of COVID-19 information** | | |
| Radio | 18 | 17.3 |
| Other social media | 31 | 29.8 |
| Official website | 21 | 20.2 |
| Friends | 3 | 2.9 |
| Family | 4 | 3.8 |
| Television | 27 | 26.0 |

**Table 2. Distribution of correct answers on baseline evaluation of knowledge and attitudes on COVID-19 among hotels' staff in Kigali (N = 104).**

| Question statement | | Answered correctly n (%) |
|---|---|---|
| **Knowledge** | | |
| Is COVID-19 fatal? | | 55 (52.9) |
| COVID-19 is thought to originate from bats? | | 54 (51.9) |
| COVID-19 signs and symptoms are? | Headache | 71 (68.3) |
| | Fever | 91 (87.5) |
| | Cough | 82 (78.8) |
| | Sore throat | 49 (47.1) |
| | Flu | 66 (63.5) |
| Disease can be transmitted during the asymptomatic phase? | | 57 (54.8) |
| COVID-19 is transmitted through? | By droplet | 5(4.8%) |
| | Physical contact <1m | 86 (82.7) |
| | Faecal-oral routes | 38 (36.5) |
| | Animals to Humans | 48(46.1) |
| **Attitudes** | | |
| Washing hands should also be applied for non-infectious customer? | | 79 (76.0) |
| Washing hands with soap & water can help in prevention of COVID-19 transmission? | | 98(94.2) |
| After touching customer surroundings. Is it always necessary to wash hand? | | 99(95.2) |
| Using hand Sanitizer is better than water and soap? | | 73(69.2) |
| Social distancing of 1m can help in prevention of COVID-19 transmission? | | 95(91.3) |
| Wearing gloves when handling customer linens can protects you from infectious diseases? | | 85(81.7) |
| During the outbreak, eating well-cooked and safely handled meat is it safe? | | 53(51.0) |
| The best container for chlorine solution is metallic recipient? | | 77(74.0) |
| Sunlight makes the chlorine solution stronger? | | 22(21.1) |
| Always carry garbage bags on your back to avoid contact with your face | | 55(52.9) |
| Is it necessary to disinfect equipment's, vehicles and working area contaminated with bleach solution? | | 88(82.7) |

than those working in the administration. On bivariate analysis, the staff with a working experience of less than one year seemed to be more knowledgeable than those having more than 11 years of experience. However, this association between working experience and the level of knowledge on COVID-19 was diluted after adjustment with other covariates in multivariate analysis.

## Discussion

This study aimed to assess the baseline knowledge and attitudes of hotels' staff regarding infection prevention and control and preparedness strategies for COVID-19 to ensure their safety and protection of guests and customers, attending hotels located in Kigali. Overall, the staff have shown satisfactory knowledge and attitudes despite the disparities among different groups. The staff with the University educational level had better knowledge than their colleagues with secondary educational level or less while those working in the administration ensured better attitudes against COVID-19 than the staff working in the front-office and housekeeping.

Our findings were similar to those found in Nigeria, which have revealed that having a University educational level was associated with improved knowledge regarding infection

**Table 3. Scores obtained by participants on knowledge and attitudes evaluation in Kigali (N = 104).**

| Outcome | Scores | | Mean score±SD | Satisfactory rate n (%) | Unsatisfactory rate n (%) |
|---|---|---|---|---|---|
| | Maximum | Minimum | | | |
| Knowledge | 12 | 1 | 6.75±2.44 | 66 (63.5%) | 38 (36.5) |
| Attitudes | 11 | 3 | 7.89 ± 1.78 | 71 (68.3%) | 33 (31.7) |

SD: standard deviation.

prevention and control measures for COVID-19 [16]. This was also echoed by another study conducted in Chine [1]. Previous authors have shown that higher educational level contributed to the development of a broader range of knowledge and skills [17], which would pave a better way of the utilization of health information. Further, reports have shown that the improved accessibility to the health information through various channels, including the internet, could enable people to change their behaviours towards the utilization of appropriate infection prevention and control measures [18].

In Rwanda, various social media platforms have been recently used to disseminate the information to prevent and control the COVID-19 transmission [19]. The better uptake of and compliance to this information would have contributed to the improved knowledge regarding the COVID-19 among the staff with higher educational level compared to others with lower educational levels.

This study has also indicated that the hotels' personnel working in the administration had better attitudes against COVID-19 than those working in the areas of food and beverages preparation and distribution, and those based in the front-office or in charge of housekeeping. This could be explained by higher educational qualifications of the administrative staff compared to others.

One strength of this study was the focus on the infection prevention and control measures for COVID-19 in critical areas like hotels, where poor compliance to these measures may lead to high transmission of the infection. Thus, our findings may have policy implications, especially regarding strengthening the education of staff working in these areas with increased risks of infection transmission. The results highlight the importance of a systematic approach in consolidating evidence needed to identify priority populations for targeted intervention in the face of resource challenges during a public health emergency.

However, this study had some limitations, including the assessment, which has solely focused on knowledge and attitudes of the hotels' staff. It would have been more interesting to also assess their actual practice when at work. In addition, this study was limited to the staff, working in hotels located in the capital, Kigali. Thus, our findings could not be generalised to the country as a whole since other hotels based outside of the capital were not represented. However, these findings provided some insights, which may inform further action to prevent and control the COVID-19 in Rwanda.

## Conclusion

As the global threat of COVID-19 continues to rise, hotels' staff based in the Rwandan capital have shown satisfactory levels of knowledge and attitudes regarding appropriate measures to prevent and control the COVID-19 during the period of rapid rise of the outbreak. However, the staff with higher educational level had better comprehensive knowledge than those with lower educational level. Similarly, the staff working in the administration have shown better attitudes towards the infection prevention and control than others. Thus, educational interventions targeting the groups with lower coverage in health information are highly needed for better prevention.

**Table 4. Analysis of sociodemographic characteristics as factors influencing knowledge and attitudes on COVID-19 among hotels' staff in Kigali.**

| Variables | Factors influencing the Knowledge | | | | | | Factors influencing the attitudes | | | | | |
|---|---|---|---|---|---|---|---|---|---|---|---|---|
| | Level of knowledge | | OR (95% CI) | p-value | AOR (95% CI) | p-value | Level of attitudes | | OR (95% CI) | p-value | AOR (95% CI) | p-value |
| | Satisfied n (%) | Unsatisfied n (%) | | | | | Satisfied n (%) | Unsatisfied n (%) | | | | |
| **Age (years)** | | | | | | | | | | | | |
| 18–29 years | 25 (37.9) | 10 (26.3) | 1 | | | | 24 (33.8) | 11 (33.3) | 1 | | | |
| 30–44 years | 36 (54.5) | 24 (63.2) | 1.6 (0.67–4.08) | 0.26 | | | 42 (59.1) | 18 (54.5) | 1.0 (0.43–2.63) | 0.88 | | |
| 45–60 years | 5 (7.6) | 2 (5.3) | 1 (0.16–6.02) | 1.00 | | | 5 (7.0) | 2 (6.1) | 1.1 (0.19–6.85) | 0.88 | | |
| > 60 years | 0 | 2 (5.3) | – | | | | | 2 (6.1) | – | | | |
| **Gender** | | | | | | | | | | | | |
| Man | 42 (63.6) | 28 (73.7) | 1 | | | | 45 (63.4) | 25 (75.8) | 1 | | | |
| Woman | 24 (36.4) | 10 (26.3) | 1.6 (0.66–3.85) | 0.29 | | | 26 (36.6) | 8 (24.2) | 1.8 (0.71–4.58) | 0.21 | | |
| **Educational level** | | | | | | | | | | | | |
| Secondary | 12 (19.7) | 16 (42.1) | 1 | | | | 15 (21.1%) | 14 (42.4) | 1 | | | |
| University | 53 (80.3) | 22 (57.9) | 2.9 (1.22–7.18) | 0.016 | 2.6 (1.07–6.58) | 0.03 | 56 (78.9%) | 19 (57.6) | 2.7 (1.12–6.73) | 0.03 | 1.8 (0.73–0.41) | 0.19 |
| **Work experience (years)** | | | | | | | | | | | | |
| >1 years | 3 (4.5%) | 6 (15.8) | 7.0 (1.18–41.35) | 0.03 | | | 6 (8.4%) | 3 (9.1%) | 0.5 (0.96–3.37) | 0.54 | | |
| 1–3 years | 18 (27.3%) | 5 (13.2) | 2.5 (0.75–8.93) | 0.13 | | | 13 (18.3%) | 10 (30.3) | 0.3 (0.93–1.48) | 0.16 | | |
| 4–10 years | 31 (47.0%) | 23 (60.5) | 0.3 (0.21–4.3) | 0.97 | | | 38 (53.5%) | 16 (48.5) | 0.6 (0.19–2.38) | 0.54 | | |
| >11 years | 14 (21.2%) | 4 (10.5) | 1 | | | | 14 (19.7%) | 4 (12.1) | 1 | | | |
| **Job category** | | | | | | | | | | | | |
| Administrative | 19 (28.8) | 5 (13.2) | 1 | | | | 23 (32.4) | 1 (3.0) | 1 | | | |
| Food and beverage | 20 (30.3) | 16 (42.1) | 3.0 (0.92–9.93) | 0.07 | | | 22 (31.0) | 14 (42.4) | 0.06 (0.01–0.56) | 0.013 | 0.08 (0.01–0.67) | 0.02 |
| Front officer | 13 (19.7) | 6 (15.79) | 1.8 (0.44–7.01) | 0.42 | | | 10 (14.1) | 9 (27.3) | 0.04 (0.01–0.53) | 0.007 | 0.05 (0.01–0.54) | 0.01 |
| Housekeeper | 14 (21.2) | 11 (28.95) | 3.0 (0.80–10.5) | 0.09 | | | 16 (22.5) | 9 (27.3) | 0.07 (0.01–0.67) | 0.02 | 0.09 (0.01–0.87) | 0.04 |
| **Official website** | 15 (22.7) | 6 (15.79) | 1 | | | | 14 (19.7) | 7 (21.2) | 1 | | | |
| Radio | 8 (12.1) | 10 (26.3) | 3.1 (0.89–11.7) | 0.09 | | | 10 (14.1) | 8 (24.2) | 0.6 (0.17–2.29) | 0.48 | | |
| Social media | 22 (33.3) | 9 (23.7) | 1.02 (0.30–3.4) | 0.97 | | | 23 (32.4) | 8 (24.2) | 1.4 (0.42–4.83) | 0.56 | | |
| Friends | 2 (3.0) | 1 (2.6) | 1.25 (0.94–16) | 0.86 | | | 2 (2.8) | 1 (3.0) | 1 (0.76–13.01) | 1.00 | | |
| Family | 1 (1.50) | 3 (7.9) | 7.5 (0.64–87.1) | 0.11 | | | 3 (4.2) | 1 (3.0) | 1.5 (0.13–17.10) | 0.74 | | |
| Television | 18 (27.3) | 9 (23.7) | 1.25 (0.36–4.30) | 0.72 | | | 19 (26.8) | 8 (24.2) | 1.1 (0.34–4.04) | 0.78 | | |

## Acknowledgments

Authors are thankful of the participants, who voluntarily accepted to complete the self-administered questionnaires. The authors are also grateful to the management of Rwanda Biomedical Centre and University of Rwanda for having granted the permission to carry out this study.

## Author Contributions

**Conceptualization:** Aphrodis Hagabimana, Jared Omolo, Angela Umutoni, Albert Ndagijimana.

**Data curation:** Aphrodis Hagabimana, Olivier Nsekuye, Albert Ndagijimana.

**Formal analysis:** Aphrodis Hagabimana, Ziad El-Khatib, Aimable Musafili, Albert Ndagijimana.

**Investigation:** Aphrodis Hagabimana, Noella Benemariya, Adeline Kabeja, Helene Balisanga, Albert Ndagijimana.

**Methodology:** Aphrodis Hagabimana, Ziad El-Khatib, Aimable Musafili, Albert Ndagijimana.

**Project administration:** Aphrodis Hagabimana, Edson Rwagasore, Adeline Kabeja.

**Software:** Aphrodis Hagabimana, Olivier Nsekuye, Albert Ndagijimana.

**Validation:** Aphrodis Hagabimana, Jared Omolo, Ziad El-Khatib, Edson Rwagasore, Angela Umutoni, Aimable Musafili, Albert Ndagijimana.

**Visualization:** Aphrodis Hagabimana, Jared Omolo, Ziad El-Khatib, Edson Rwagasore, Adeline Kabeja, Angela Umutoni, Aimable Musafili, Albert Ndagijimana.

**Writing – original draft:** Aphrodis Hagabimana.

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
