## [Decision Letter · Decision Letter 0]

19 Oct 2021

PONE-D-21-30529Baseline knowledge and attitudes of COVID-19 among hotel staff.  A cross sectional study in Kigali, Rwanda.PLOS ONE

Dear Dr. Hagabimana,

Thank you for submitting your manuscript to PLOS ONE. After careful consideration, we feel that it has merit but does not fully meet PLOS ONE’s publication criteria as it currently stands. Therefore, we invite you to submit a revised version of the manuscript that addresses the points raised during the review process.

We look forward to receiving your revised manuscript.

Kind regards,

Sanjay Kumar Singh Patel, Ph.D.

Academic Editor

PLOS ONE

Journal Requirements:

2. Please amend the manuscript submission data (via Edit Submission) to include authors Jared Omolo1, Ziad El-Khatib, Edson Rwagasore, Noella Benemariya, Olivier Nsekuye, Adeline Kabeja, Helene Balisanga, Angela Umutoni and Albert Ndagijimana1.

3. We note you have included a table to which you do not refer in the text of your manuscript. Please ensure that you refer to Table 3 in your text; if accepted, production will need this reference to link the reader to the Table.

Reviewers' comments:

Reviewer's Responses to Questions

**Comments to the Author**

1. Is the manuscript technically sound, and do the data support the conclusions?

Reviewer #1: Yes

Reviewer #2: Yes

2. Has the statistical analysis been performed appropriately and rigorously? 

Reviewer #1: Yes

Reviewer #2: Yes

3. Have the authors made all data underlying the findings in their manuscript fully available?

Reviewer #1: Yes

Reviewer #2: Yes

4. Is the manuscript presented in an intelligible fashion and written in standard English?

Reviewer #1: Yes

Reviewer #2: Yes

5. Review Comments to the Author

Reviewer #1: Hotels and other pandemic establishments are associated with higher transmission of coronavirus. So, better sensitization of staff and strengthening is required to control the coronavirus pandemic. In this paper, the authors investigate the baseline knowledge and attitudes on COVID-19 among hotel staff in Kigali, Rwanda. A cross-sectional study is performed using structured questionnaires to 104 participants. Although the number of participants is small, it provides an essential understanding of the COVID-19 pandemic.

Minor comments:

1) Please discuss how the results stated in the manuscript are significant with only 104 hotel staff. Also, state the reason why more samples are not included in the manuscript.

2) The manuscript is well written. However, there are typos in the manuscript, such as fetal should be correct to fatal. Please go through the manuscript and correct all spelling errors.

3) In the discussion section, please discuss how present studies is relevant in the present and future pandemic.

4) There are some formatting errors in the reference section. Please correct them.

Reviewer #2: The manuscript by Hagabimana et al. “Baseline knowledge and attitudes of COVID-19 among hotel staff. A cross sectional study in Kigali, Rwanda.” requires revision to address major concerns.

Comments

1. The manuscript may be polished extensively for the English language.

2. Introduction (first paragraph): The information about mortality rate and various prevention approaches should be provided related to immunity and health i.e. doi: 10.1007/s12088-020-00908-0.

3. Introduction, the importance of this study may be more specifically highlighted and justified.

4. The author should provide at least one or two illustrations (additional Figures) to highlight the summary and significance.

---

## [Author Response · Author response to Decision Letter 0]

6 Dec 2021

Reviewer #1:

Hotels and other pandemic establishments are associated with higher transmission of coronavirus. So, better sensitization of staff and strengthening is required to control the coronavirus pandemic. In this paper, the authors investigate the baseline knowledge and attitudes on COVID-19 among hotels’ staff in Kigali, Rwanda: a cross-sectional study is performed using structured questionnaires to 104 participants. Although the number of participants is small, it provides an essential understanding of the COVID-19 pandemic.

Authors ‘reply: We would like to thank Reviewer 1 for the overall summary and appreciation of the study in terms of an essential understanding of the COVID-19 pandemic.

Minor comments:

1) Please discuss how the results stated in the manuscript are significant with only 104 hotel staff. Also, state the reason why more samples are not included in the manuscript.

Authors’ reply: To the best of our knowledge, this was the first study that was conducted in the domain of hotel industry in urban area of a low-income country. We believe that this study provided important insights about the infection and control measures for COVID-19 in the hospitality sector. However, there were some limitations, which prevented us to conduct a larger survey among hotels’ staff in Kigali and other parts of the country. These limitations were related to the lack of study funding and movements’ restrictions during the time of the outbreak of COVID-19 pandemic. However, even though our findings may not provide a countrywide picture regarding the infection prevention and control measures for COVID-19 among hotels’ staff, we do believe that they provided some illumination on the importance of a systematic approach in consolidating evidence needed to identify priority populations for targeted intervention.

2) The manuscript is well written. However, there are typos in the manuscript, such as fetal should be correct to fatal. Please go through the manuscript and correct all spelling errors.

Authors’ reply: Thank you for pointing this out. We have revised the manuscript and addressed typos errors, as suggested. 

3) In the discussion section, please discuss how present studies is relevant in the present and future pandemic.

Authors’ reply: Thank you for pointing this out. We have revised the discussion, as suggested. 

4) There are some formatting errors in the reference section. Please correct them.

Authors’ reply: Thank you for pointing this out. We have corrected errors in the reference section.

Reviewer #2:

The manuscript by Hagabimana et al. “Baseline knowledge and attitudes of COVID-19 among hotel staff. A cross sectional study in Kigali, Rwanda.” requires revision to address major concerns.

Authors’ reply: We thank the esteemed Reviewer 2 for considering our paper and we hope you will find our updated version meeting your expectations.

Comments and answers:

1. The manuscript may be polished extensively for the English language.

Authors’ reply: Thank you for pointing this out. In the revised version of the manuscript, we have addressed the English language.

2. Introduction (first paragraph): The information about mortality rate and various prevention approaches should be provided related to immunity and health i.e. doi: 10.1007/s12088-020-00908-0.

Authors’ reply: We searched for the suggested reference, using the above doi, by the reviewer 2. The above doi did lead us to the reference: Rishi et.al.; Diet, Gut Microbiota and COVID-19; Indian J Microbiol; Sep 2020; https://pubmed.ncbi.nlm.nih.gov/33012868/. We feel this reference is out of scope in relation to our paper. We ask kindly reviewer 2 to confirm the reference.

3. Introduction, the importance of this study may be more specifically highlighted and justified.

Authors’ reply: Thank you for pointing this out. We have considered this comment in our revised manuscript.

4. The author should provide at least one or two illustrations (additional Figures) to highlight the summary and significance.

Authors’ reply: Thank you for pointing this out. This comment was addressed accordingly in the current version of the manuscript.

---

## [Editor Report · Decision Letter 1]

9 Dec 2021

Baseline knowledge and attitudes of COVID-19 among hotel staff.  A cross sectional study in Kigali, Rwanda.

PONE-D-21-30529R1

Dear Dr. Hagabimana,

We’re pleased to inform you that your manuscript has been judged scientifically suitable for publication and will be formally accepted for publication once it meets all outstanding technical requirements.

Kind regards,

Sanjay Kumar Singh Patel, Ph.D.

Academic Editor

PLOS ONE

---

## [Editor Report · Acceptance letter]

20 Dec 2021

PONE-D-21-30529R1 

Baseline knowledge and attitudes on COVID-19 among hotels’ staff: a cross-sectional study in Kigali, Rwanda. 

Dear Dr. Hagabimana:

I'm pleased to inform you that your manuscript has been deemed suitable for publication in PLOS ONE. Congratulations! Your manuscript is now with our production department. 

Kind regards, 

on behalf of

Dr. Sanjay Kumar Singh Patel 

Academic Editor

PLOS ONE